

# MTBseq: a comprehensive pipeline for whole genome sequence analysis of *Mycobacterium tuberculosis* complex isolates

Thomas Andreas Kohl[1,*], Christian Utpatel[1,*], Viola Schleusener[1], Maria Rosaria De Filippo[2], Patrick Beckert[1,3], Daniela Maria Cirillo[2] and Stefan Niemann[1,3]

[1] Molecular and Experimental Mycobacteriology, Research Center Borstel, Borstel, Germany
[2] Emerging Bacterial Pathogens Unit, Division of Immunology, Transplantation and Infectious Diseases, IRCCS San Raffaele Scientific Institute, Milan, Italy
[3] German Center for Infection Research (DZIF), partner site Hamburg—Lübeck—Borstel—Riems, Borstel, Germany
[*] These authors contributed equally to this work.

## ABSTRACT

Analyzing whole-genome sequencing data of *Mycobacterium tuberculosis* complex (MTBC) isolates in a standardized workflow enables both comprehensive antibiotic resistance profiling and outbreak surveillance with highest resolution up to the identification of recent transmission chains. Here, we present MTBseq, a bioinformatics pipeline for next-generation genome sequence data analysis of MTBC isolates. Employing a reference mapping based workflow, MTBseq reports detected variant positions annotated with known association to antibiotic resistance and performs a lineage classification based on phylogenetic single nucleotide polymorphisms (SNPs). When comparing multiple datasets, MTBseq provides a joint list of variants and a FASTA alignment of SNP positions for use in phylogenomic analysis, and identifies groups of related isolates. The pipeline is customizable, expandable and can be used on a desktop computer or laptop without any internet connection, ensuring mobile usage and data security. MTBseq and accompanying documentation is available from https://github.com/ngs-fzb/MTBseq_source.

Corresponding author
Stefan Niemann,
sniemann@fz-borstel.de

## INTRODUCTION

The recent development of next-generation sequencing (NGS) technologies in line with the reduction of sequencing costs and introduction of benchtop instruments allows the use of whole-genome sequencing (WGS) as routine tool for bacterial strain characterization, for example, for resistance prediction and in-depth genotyping of bacterial isolates (*Walker et al., 2017*). This development has led to significant

improvements for the epidemiological surveillance of major pathogens such as the *Mycobacterium tuberculosis* complex (MTBC), the causative agent of tuberculosis (TB) (*Dheda et al., 2017*; *Merker et al., 2017*; *Walker et al., 2018*; *Zignol et al., 2018*). With 10 million new cases in 2017 and the emergence of multidrug resistant strains, TB remains one of the 10 leading causes of death worldwide (*World Health Organization, 2018*).

The application of WGS technologies clearly advances resistance prediction, outbreak detection, and genomic surveillance of MTBC (*Merker et al., 2017*). At the same time, no comprehensive pipeline allowing for the full analysis of individual datasets and a set of samples has been proposed so far. Comprehensive and powerful open source packages for standardized NGS analysis exist such as UGENE (*Okonechnikov, Golosova & Fursov, 2012*), The Galaxy Project (*Afgan et al., 2016*), or GenePattern (*Reich et al., 2006*), as well as programming language toolkits such as BioPerl (*Stajich et al., 2002*), BioPython (*Cock et al., 2009*), or BioRuby (*Goto et al., 2010*). However, to set up fully integrated workflows from scratch that allow for an accurate and meaningful analysis of NGS data from clinical MTBC strains, still requires programming expertise, and trained bioinformatics personnel. This constrains the application of NGS analysis to specialized laboratories, leads to a huge diversity of analysis pipelines with group specific solutions and seriously complicates comparison of results. Several automated pipelines for resistance determination have been developed, namely three web services (CASTB (*Iwai et al., 2015*), PhyResSE (*Feuerriegel et al., 2015*), and TBProfiler (*Coll et al., 2015*)), and two local software solutions (KvarQ (*Steiner et al., 2014*) and Mykrobe Predictor TB (*Bradley et al., 2015*)). All these tools enable non-specialists to infer drug resistance from WGS data of MTBC strains and also provide phylogenetic classification results. Their respective strengths and weaknesses have been compared before (*Schleusener et al., 2017*).

Still, bioinformatics data analysis is a clear bottleneck that restricts accessibility and wide adoption of NGS technologies in TB research, diagnostics and surveillance.

To address this challenge, we developed MTBseq, an automated pipeline for MTBC NGS data analysis. MTBseq combines all necessary steps for the analysis of NGS datasets from MTBC strains ranging from basic analysis procedures such as mapping of reads to the reference genome, to detection of variant positions annotated with known association to antibiotic resistance, and lineage classification based on phylogenetic single nucleotide polymorphisms (SNPs). In addition, MTBseq enables the comparative analysis of multiple samples to produce joint lists of variants and a FASTA alignment of SNP positions for use with tree-based approaches.

## MATERIALS AND METHODS
### Description of MTBseq individual steps
MTBseq employs the widely used open source programs BWA (*Li & Durbin, 2009*), SAMtools (*Li et al., 2009*), PICARD-tools (https://broadinstitute.github.io/picard/) and Genome Analysis Toolkit (GATK) (*McKenna et al., 2010*). The workflow starts with raw FASTQ formatted sequences (reads). Within the mapping step, the BWA-MEM

algorithm and SAMtools are used for a generalized mapping procedure (TBbwa). The output is a deduplicated and a sorted alignment file saved in the binary alignment format BAM (*Li et al., 2009*). Next, GATK is used for base call recalibration and realignment of reads around insertions or deletions (TBrefine). Variant calling is a multi-step process that starts with the SAMtools mpileup utility, providing coverage information at single base resolution (TBpile) and employs thresholds for coverage and base quality (TBlist). With default settings, variants need to be indicated by four reads mapped in each forward and reverse orientation, respectively, at 75% allele frequency, and by at least four calls with a phred score of at least 20. Therefore, MTBseq will report reliably detected variants if they are present in about 75% or more of the bacterial population when using default values. In the sample specific report file, detected SNPs and insertions or deletions are annotated with respective metadata, including resulting amino acid changes for SNPs in coding regions and association to antibiotic resistance (TBvariants). The sixth step creates a descriptive statistics report of the mapping and variant detection steps, giving a clear indication of overall dataset performance (TBstats). The last sample specific module enables the phylogenetic classification of the input sample(s) according to phylogenetically informative SNPs from the literature (*Coll et al., 2014*; *Homolka et al., 2012*; *Merker et al., 2015*) (TBstrains). A comparative analysis of multiple samples is provided by the TBjoin and TBamend modules, which can be executed for any set of samples already processed by the sample specific workflow. The primary result is a list, containing information of all positions for which a variant had been detected in any of the input samples. To facilitate phylogenetic analysis, variant subsets are automatically generated, filtered for repetitive regions (*Comas et al., 2010*) and resistance-associated genes, the kind of variant detected, and the presence of other variants within a window of 12 bp within the same dataset (*Walker et al., 2013*). In addition, FASTA formatted sequences are generated as direct input for targeted applications (e.g., tree reconstruction algorithms). The comparative analysis finishes with grouping input samples according to the number of distinct SNP positions (TBgroups).

Although MTBseq offers a batch mode, the described modules are functionally separated and independent in execution (Fig. 1, blue boxes). This architecture allows every single module to be executed directly by the user and ensures expandability of the workflow by developers. A simple checkpoint system is implemented between every step (Fig. 1, blue lines with diamond symbol), keeping track of analysis results already generated. Therefore, the pipeline parts are executed only for samples that have the respective input and lack the respective output of the module invoked. In order to ensure this functionality, MTBseq creates its own working environment at the location of execution. With MTBseq, we aim to provide a sequence analysis pipeline for the MTBC that is customizable, expandable, user friendly, and standardized. At the same time, we aim to offer users the opportunity to customize the functionality to their specific needs. Therefore, all parameters are set to default values while especially thresholds within variant calling and in the comparative part of the program can be easily modified by the user. In this way, we want to enable any user with basic Linux experience to run the software,

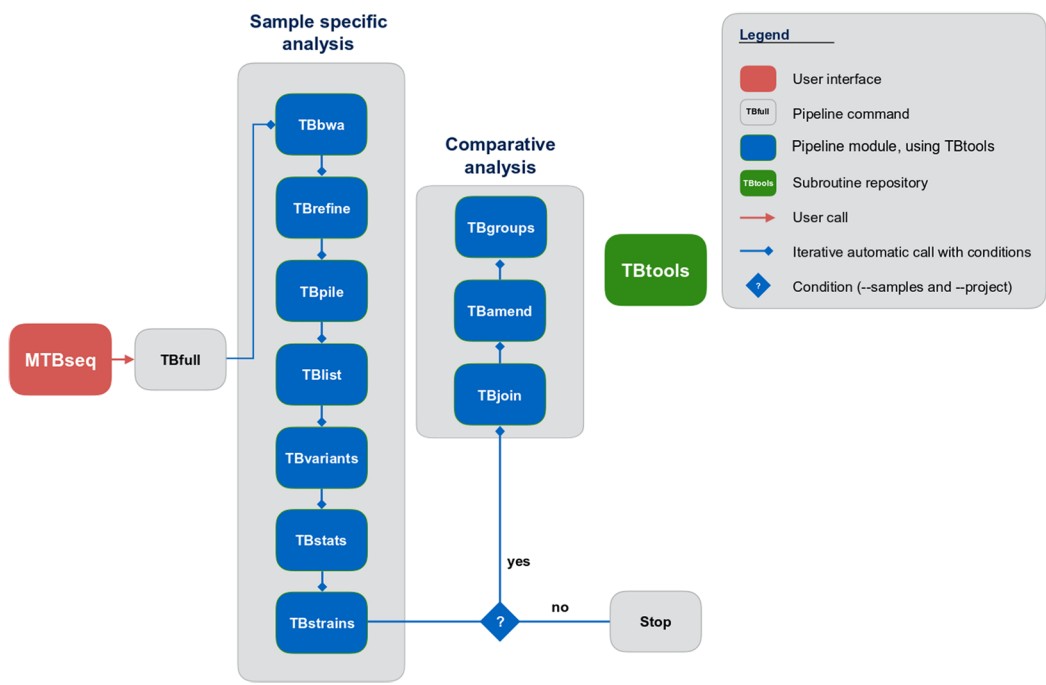

**Figure 1 Schematic representation of the MTBseq workflow.** Modules encapsulating specific functionality shown in blue boxes.

while also allowing for the easy modification of the functionality by users with a bioinformatics background.

MTBseq uses the *M. tuberculosis* H37Rv genome (NC_000962.3) and corresponding metadata as reference by default but the pipeline can be used with any MTBC or non-MTBC bacterial reference genome and corresponding annotation, supplied by the user. For base quality recalibration and annotation of resistance associated or phylogenetic variants, a default list is provided with MTBseq, which can be easily replaced by the user with a respective compilation, for example, as drawn from the ReSeqTB or CRyPTIC initiatives. This is especially important since this data forms the basis for inferring a resistance profile from the detected variants.

## Programming language and availability

MTBseq is written in the Perl programming language and can be obtained from https://github.com/ngs-fzb/MTBseq_source, including the full source code, documentation, and usage guidelines.

## RESULTS

### General workflow of MTBseq

MTBseq consists of two workflows with seven modules encompassing a sample specific analysis and three modules enabling comparative analysis of multiple datasets (Fig. 1). The sample specific workflow comprises read alignment to the *M. tuberculosis* H37Rv reference genome (NC_000962.3), refining the resulting alignment, and the detection of SNPs, as well as insertions and deletions. Reported variants are annotated with

applicable metadata, in particular whether an association with antibiotics resistance is known. In addition, MTBseq performs a phylogenetic classification using a set of informative SNPs (*Coll et al., 2014*; *Homolka et al., 2012*; *Merker et al., 2015*).

For multiple datasets, the pipeline enables a comparative analysis of a set of samples (Fig. 1), which includes an agglomerative clustering (e.g., inference of transmission groups from pairwise distances) and the determination of informative positions for the reconstruction of phylogenetic trees. MTBseq can be executed in a batch mode without any user intervention. This is an important issue, if large numbers of datasets have to be analyzed. Details of individual steps are detailed in the Materials and Methods section. Importantly, the pipeline can be run nearly completely automated from one command line call, with all parameters pre-set to appropriate default values.

## Antibiotic resistance detection and classification

Using procedures similar to a recently published study (*Schleusener et al., 2017*), we performed a systematic evaluation of MTBseq for the prediction of antibiotic resistance against the four first line drugs in TB therapy (isoniazid, rifampicin, ethambutol, and pyrazinamide) and streptomycin, as well as for phylogenetic classification of MTBC isolates. The dataset used consisted of 91-well characterized strains of a collection from Sierra Leone, for which both WGS (ENA accession number PRJEB7727) and Sanger sequencing data was available (*Schleusener et al., 2017*). We compared the results with five other software solutions available for resistance inference and phylogenetic classification of MTBC datasets, that is, CASTB (*Iwai et al., 2015*), KvarQ (*Steiner et al., 2014*), Mykrobe Predictor TB (*Bradley et al., 2015*), PhyResSE (*Feuerriegel et al., 2015*), and TBProfiler (*Coll et al., 2015*). Overall, MTBseq, PhyResSE, and TBProfiler exhibited the highest sensitivity and specificity among the tools tested (Table 1).

Using MTBseq, we obtained a 100% sensitivity compared to Sanger sequencing results for rifampicin and isoniazid, the most important drugs for TB treatment. For ethambutol, streptomycin, and pyrazinamide, the sensitivity was also 100%, and specificity at least 90% (Table 1). In these calculations, we included the detection of insertions or deletions in genes annotated by MTBseq as resistance associated (Table S1) and the detection of resistant subpopulations, which MTBseq reported correctly when parameters were adjusted to detect low frequency variants (Table S1). In the low frequency detection mode (set with the "–lowfreq_vars" option), MTBseq will consider the majority allele different from the wild type base. Regarding the classification of samples into known phylogenetic lineages, MTBseq was able to classify 11 strains not resolved by classical genotyping (classical genotyping result: "none"; Table S1). As by MTBseq default settings, a set of known variant positions was used for base quality calibration (Table S2).

## Phylogenetic analysis and cluster detection

We used a well characterized dataset of 26 isolates from an outbreak in Hamburg to perform a phylogenetic analysis with MTBseq (*Kohl et al., 2014*). It contains NGS data from 26 isolates from a longitudinal population-based study in Hamburg, which were recognized as belonging to a potential outbreak as all isolates presented with the same

**Table 1 Sensitivity and specificity for resistance prediction of different tools.**

| Antibiotic | Sanger #R | Sanger #S | CASTB Sens | CASTB Spec | PhyResSE Sens | PhyResSE Spec | KvarQ Sens | KvarQ Spec | Mykrobe Predictor TB Sens | Mykrobe Predictor TB Spec | TBProfiler Sens | TBProfiler Spec | MTBseq Sens | MTBseq Spec |
|---|---|---|---|---|---|---|---|---|---|---|---|---|---|---|
| INH | 28 | 63 | 89 (72, 98) | 100 (92, 100) | 100 (82, 100) | 98 (91, 100) | 86 (67, 96) | 100 (92, 100) | 89 (72, 98) | 100 (92, 100) | 93 (76, 99) | 84 (73, 92) | 100 (82, 100) | 98 (91, 100) |
| RMP | 18 | 73 | 94 (73, 100) | 100 (93, 100) | 100 (74, 100) | 99 (93, 100) | 94 (73, 100) | 100 (93, 100) | 100 (74, 100) | 99 (93, 100) | 100 (74, 100) | 99 (93, 100) | 100 (74, 100) | 100 (93, 100) |
| SM | 37 | 54 | 30 (16, 47) | 100 (90, 100) | 100 (86, 100) | 100 (90, 100) | 57 (39, 73) | 100 (90, 100) | 57 (39, 73) | 100 (90, 100) | 57 (39, 73) | 100 (90, 100) | 100 (86, 100) | 100 (90, 100) |
| EMB | 15 | 76 | 53 (27, 79) | 100 (93, 100) | 94 (70, 100) | 100 (93, 100) | 53 (27, 79) | 100 (93, 100) | 47 (21, 73) | 99 (93, 100) | 94 (70, 100) | 99 (93, 100) | 100 (71, 100) | 100 (93, 100) |
| PZA | 11 | 80 | 45 (17, 77) | 100 (93, 100) | 100 (62, 100) | 99 (93, 100) | 45 (17, 77) | 100 (93, 100) | n.a. | n.a. | 100 (62, 100) | 99 (93, 100) | 100 (62, 100) | 99 (93, 100) |

**Notes:**

Evaluation of resistance deduction from whole-genome sequence data by programs CASTB, PhyResSE, KvarQ, Mykrobe Predictor TB, TBProfiler, and MTBseq, with sensitivity (Sens) and specificity (Spec) estimated with 95% confidence intervals compared to Sanger sequencing results (#R resistant, #S sensitive).
INH, isoniazid; RMP, rifampicin; SM, streptomycin; EMB, ethambutol; PZA, pyrazinamide.

classical genotyping patterns. All 26 isolates were analyzed with MTBseq in a joint comparison using parameters set to default values, with 4,304,720 out of the 4,411,532 bp of the H37Rv reference genome fulfilling the thresholds for variant detection. In total, 988 SNP positions were identified for a phylogenomic analysis, and the FASTA file produced by MTBseq was used as input for the tree construction program FastTree 2 (*Price, Dehal & Arkin, 2010*).

In addition, MTBseq was configured to detect clustered isolates with a threshold of 12 bp to the nearest cluster member. Both the constructed tree and the clusters indicate a central group of 22 isolates forming a tight cluster, and four isolates (1024-01, 3929-10, 6631-04, 6821-03) not related to this outbreak (Fig. 2). The same can be seen in the matrix of pairwise distances (Fig. 3).

## DISCUSSION

We developed MTBseq to overcome constraints in the bioinformatics analysis of NGS data from clinical MTBC strains and to provide a standard analysis pipeline to increase the accessibility and adoption of NGS technologies in TB research, diagnostics, and surveillance.

MTBseq employs a reference mapping based workflow, reports detected variant positions annotated with known association to antibiotic resistance and performs a lineage classification based on phylogenetic SNPs. In the joint analysis of multiple datasets, MTBseq provides a joint list of variants, a SNP distance matrix, a FASTA alignment of SNP positions for use in phylogenomics, and identifies groups of related isolates.

Here, we demonstrated the sensitivity and accuracy for resistance profiling and genotyping of MTBseq. Compared to Sanger sequencing results we obtained a 100% sensitivity for rifampicin, isoniazid, ethambutol, streptomycin, and pyrazinamide with the specificity being at least 90% for the latter four. For the correct detection of resistance-associated minority variants the low frequency detection mode of MTBseq had to be employed in which the majority allele other than wild type is considered per position. This mode should preferably be used for detection of resistance-associated variants if resistant subpopulations are suspected. This could be the case during early stages of drug-resistance acquisition or mixed TB infections. Here, it is important to keep in mind that due to the lack of an accessible gold standard sensitivity of specificity of the low frequency mode have not yet been evaluated. The phylogenetic classification into lineages and sublineages of the MTBC by MTBseq was in overall concordance with results from traditional genotyping and on par with PhyResSE and TBProfiler.

The results of the phylogenetic analysis and cluster detection of 26 isolates from an outbreak in Hamburg are in full agreement with published findings for this dataset (*Kohl et al., 2014*), which also identified a central cluster of 22 isolates and the same four outlying isolates. For tree construction, we chose the program FastTree 2, but as MTBseq provides a full FASTA alignment of SNP positions for phylogenetic analysis any phylogenomic suite can be used.

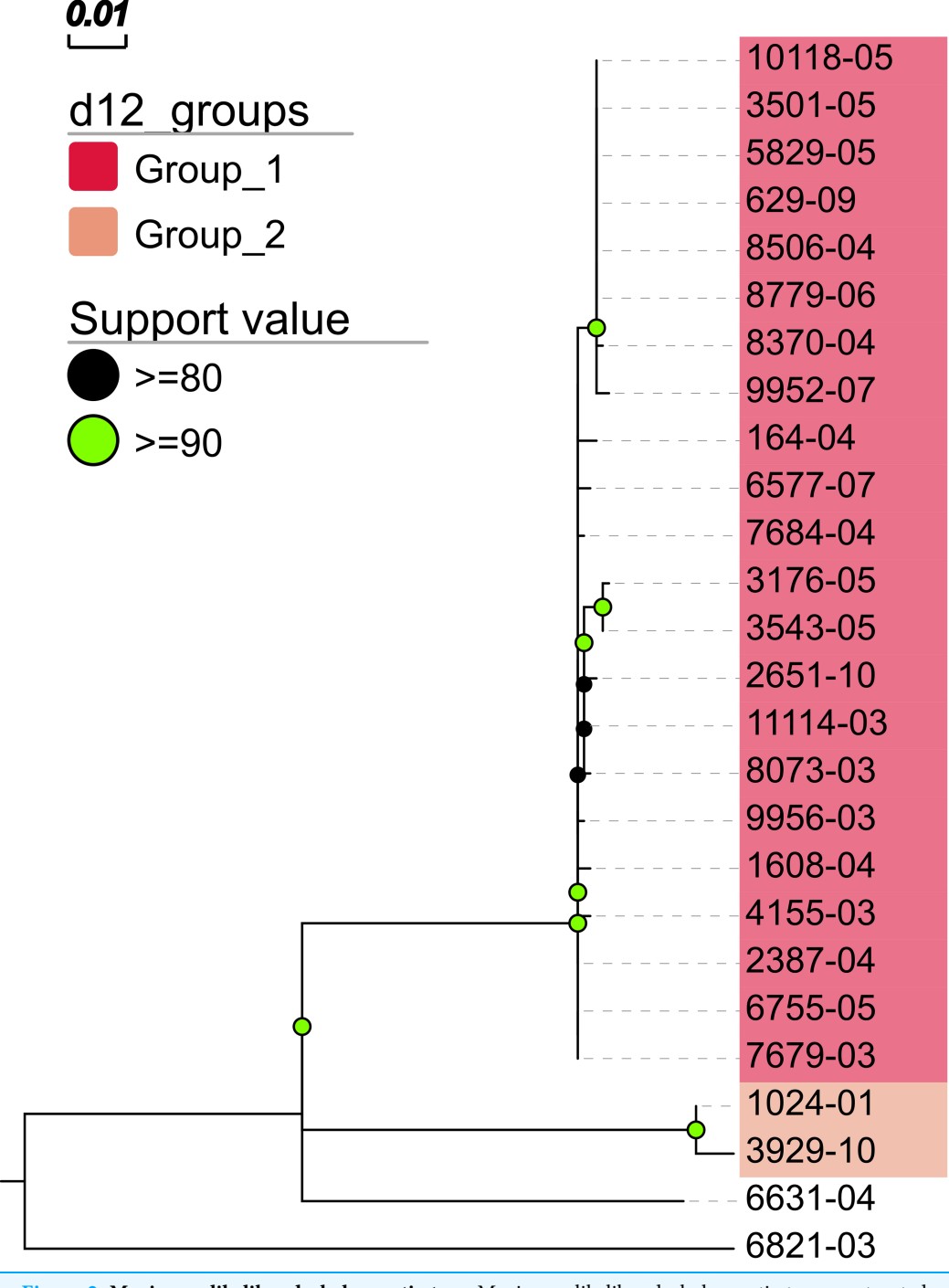

**Figure 2 Maximum likelihood phylogenetic tree.** Maximum likelihood phylogenetic tree constructed from the aligned set of SNP positions determined by MTBseq from a collection of 26 MTBC isolates suspected to form an outbreak (*Kohl et al., 2014*). For tree construction, we employed the program FastTree version 2 (*Price, Dehal & Arkin, 2010*) in the double precision built with a general time reversible (GTR) substitution model, 1,000 resamples, and Gamma20 likelihood optimization. The resulting tree was visualized with the FigTree and EvolView (*He et al., 2016*) tools. d12_groups: Groups of clustered isolates were determined by MTBseq with a maximum distance threshold of 12 SNPs using single-linkage clustering and the detected groups are indicated by the colored sample labels. Support value: Reliability values for splits based on resampling over 80% are shown.

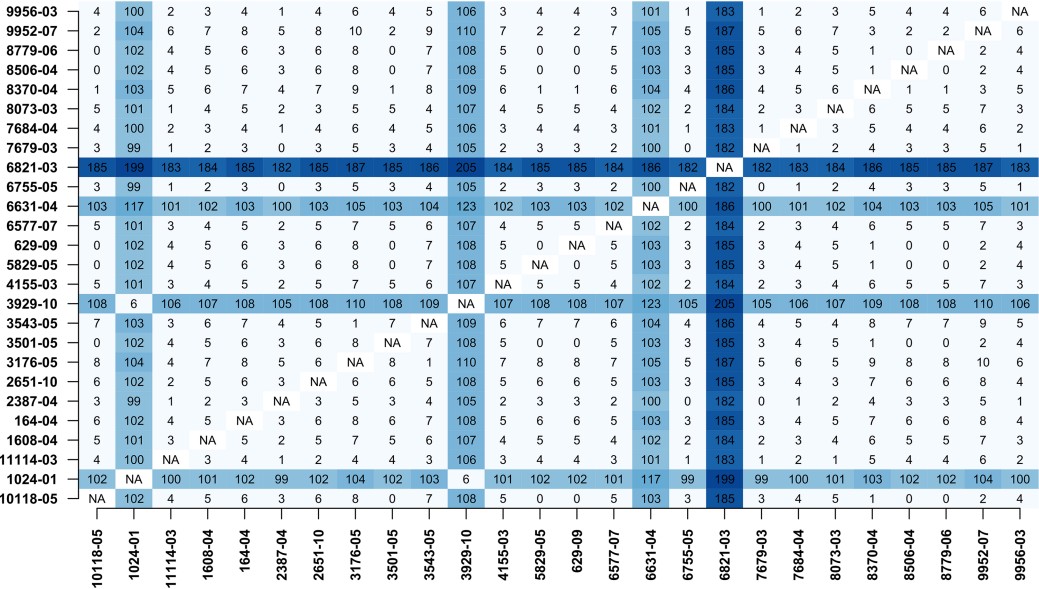

**Figure 3 Pairwise distance matrix.** Pairwise distance matrix calculated by MTBseq from a set of 26 MTBC isolates with identical traditional genotyping patterns suspected to form an outbreak (*Kohl et al., 2014*). The distance between samples is calculated from the detected variants and smaller distances indicate more closely related samples. Out of the 26 isolates, 22 have overall small pairwise distances indicative of a common cluster. The respective entries for the four remaining isolates are marked in blue (1024-01, 3929-10, 6631-04, 6821-03).

# CONCLUSIONS

MTBseq provides a comprehensive analysis pipeline for WGS analysis of MTBC NGS data. The pipeline is fully customizable and the functionality can be easily adjusted and extended, both by modifying the implementation and by adding further modules to the respective workflows. At the same time, the pipeline can be run nearly completely automated from one command line call, with all parameters pre-set to appropriate default values. We demonstrated the accuracy and sensitivity for resistance profiling, genotyping, and comparative analysis, concluding that MTBseq is a suitable automated solution for resistance deduction, and phylogenetic classification and analysis of MTBC whole genome datasets. The full source code with accompanying documentation and usage guidelines is provided at https://github.com/ngs-fzb/MTBseq_source. MTBseq thus provides a full automatized analysis pipeline for NGS datasets from MTBC strains, further paving the way for efficient application of WGS in the characterization of bacterial pathogens.

# ACKNOWLEDGEMENTS

The authors thank Robin Koch, Alexandra Dangel, Conor Meehan, and Matthias Merker for their thorough testing of the MTBseq package and helpful input during development.

### Funding

Parts of this work were funded by the European Community's Seventh Framework Program (FP7/2007-2013) under grant agreement 278864 in the framework of the Patho-NGen-Trace project and the German Center for Infection Research (DZIF). The publication of this article was funded by the Open Access Fund of the Leibniz Association. The funders had no role in study design, data collection and analysis, decision to publish, or preparation of the manuscript.

### Grant Disclosures

The following grant information was disclosed by the authors:
European Community's Seventh Framework Program: FP7/2007-2013.
German Center for Infection Research: DZIF.

### Competing Interests

The authors declare that they have no competing interests.

### Author Contributions

- Thomas Andreas Kohl conceived and designed the experiments, performed the experiments, analyzed the data, prepared figures and/or tables, authored or reviewed drafts of the paper, approved the final draft.
- Christian Utpatel conceived and designed the experiments, performed the experiments, analyzed the data, prepared figures and/or tables, authored or reviewed drafts of the paper, approved the final draft.
- Viola Schleusener performed the experiments, analyzed the data, prepared figures and/or tables, authored or reviewed drafts of the paper, approved the final draft.
- Maria Rosaria De Filippo performed the experiments, analyzed the data, authored or reviewed drafts of the paper, approved the final draft.
- Patrick Beckert performed the experiments, analyzed the data, authored or reviewed drafts of the paper, approved the final draft.
- Daniela Maria Cirillo conceived and designed the experiments, contributed reagents/materials/analysis tools, authored or reviewed drafts of the paper, approved the final draft.
- Stefan Niemann conceived and designed the experiments, contributed reagents/materials/analysis tools, authored or reviewed drafts of the paper, approved the final draft.

### Data Availability

   GitHub: https://github.com/ngs-fzb/MTBseq_source.

### Supplemental Information

Supplemental information for this article can be found online at http://dx.doi.org/10.7717/peerj.5895#supplemental-information.

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
