# Peer review of "MTBseq: a comprehensive pipeline for whole genome sequence analysis of Mycobacterium tuberculosis complex isolates"

_PeerJ, doi:10.7717/peerj.5895_

## Round 0.1 · original submission · Major Revisions

Your manuscript has been thoroughly assessed. While reviewers were supportive of your tool, they have highlighted a number of issues that I ask you to carefully and comprehensively address.

Reviewer 1 ·

Basic reporting

No comment.

Experimental design

Authors use their pipeline to identify SNPs associated with drug resistance and present a table with sensitivities and specificities. While it is absolutely helpful to see how MTBSeq compares to similar tools/pipelines, I note that Sanger sequencing is used as the gold standard, instead of phenotypic DST. If the aim is to provide information on accuracy for detecting drug resistance, DST would be the appropriate gold standard. Please show the calculations with these data as the gold standard, in addition to the current table (as this is helpful to assess the accuracy of SNP calls made using the various pipelines).

Authors recommend using the ‘low-freq’ option to detect minority resistance alleles. Would it be possible to provide some information on the accuracy of this mode for detecting these minority alleles, especially as the authors recommend using it for detection of all resistance-associated variants (lines 205-207)? The numbers might be too small using this dataset; have authors tested on any other datasets with drug resistance?

The authors state they include base quality recalibration in their pipeline, but what positions were used for this? From authors’ previous work (and reviewing the manual on git), these seem to be known resistance-associated mutations, but a supplementary table with these sites listed should be included in the manuscript. This is also important to include, as the reader should know which resistance-associated mutations are being investigated with MTBseq.

Are authors excluding variants within the PE/PPE regions? If so, please indicate this in the manuscript and provide a table listing the positions that are automatically masked.

Line 179 – ‘joint comparison’ – do authors mean they did multi-sample SNP calling was done? Or was this single calling, with VCFs merged afterwards? Please clarify.

Validity of the findings

I note in the Supplementary Table that pncA mutations (it is not indicated which one, so please specify) were reported in low-freq mode for MTBseq, but reported as consensus calls for TBprofiler and PhyResSE. It would be helpful to know what differences in the pipeline were responsible for this, as this suggests the alternative allele was the majority in other pipelines and somehow only minority in MTBseq.

On line 208, authors state their phylogenetic tree is on par with that of TBprofiler and PhyResSE. First, could authors please put these trees in the supplement so this can be assessed by the reader? Also, did both of those tools also use FastTree2, along with the same model of nucleotide substitution? If comparing trees (constructed with the fasta from the author’s pipeline), the same tool and version should be used across all tools.

Figure 2 – As a bootstrap support of 80% is often used to assess confidence, it would be useful to have this indicated instead of 70%.

Additional comments

This pipeline has the potential to be very useful. I really like that it can be run overall and also by module. However, authors state the aim of this pipeline is to assist non-bioinformaticians and those without programming expertise, but this is definitely not ready for such users. One key issue is that the pipeline has only been tested on Linux and no instructions are provided for installation on any other OS, other than to tell the user to set up a virtual machine. This is something that absolutely needs to be addressed if the authors want this pipeline to be used readily. Average clinical microbiologists (and even some bioinformaticians) do not work on a Linux OS. At the least, please set up clear instructions and test this pipeline on a Mac OS.

Regarding the default settings, it would also be helpful to have these stated explicitly for each tool (e.g., the defaults for BWA are not provided). It would also be useful to have a summary table of all the filters that are applied. For example, I didn’t see any filtering on mapping quality or removing reads that map ambiguously, or filtering on allele frequency?

If the pipeline allows users to modify default settings, such that a variety of different thresholds can be used, is it really going to allow for ‘standardized’ analysis across studies? Perhaps useful to elaborate a little on this point.

Minor point - would it be possible to remove the requirement for ‘library’ in the naming scheme or add some flexibility to this? This is not standard naming protocol for many institutions or sequencing centers.

Reviewer 2 ·

Basic reporting

### Clear and unambiguous, professional English used throughout.
The article submitted by Kohl et al. for the most part is written well. There are quite a few instances throughout the manuscript where typos were found or sentences were constructed in a confusing manner/ hard to interpret on the first read through. Please address the highlighted sentences/grammar mistakes in the returned review manuscript. Some examples are extra "s" at ends of words, incorrect use of conjunctions or the omission of fullstops.
Furthermore, line 82-88: All one sentence! Rephrase for reading ease.


### Literature references, sufficient field background/context provided.
One of the big flags is the lack of proper referencing, especially for the tools against which MTBseq was benchmarked. Prime example on line 160.
Furthermore, there is no mention of KvarQ, mykrobe, TBprofiler etc. in the introduction. These should be mentioned as they were benchmarked as part of the work presented.

Line 78: "....makes comparison of results virtually impossible". This statement is too strong, please rephrase. Results may not be identical between labs/groups but results are comparable and generally agree for the most part.

Line 69-70: "....no standard analysis pipeline....proposed so far" ....What about TBprofiler?? http://tbdr.lshtm.ac.uk/

Line 149: Suggest phrase change from 'critical issue'.


### Figures relevant, high quality, well labelled and described
Fig1:
Nice and easy to understand.

Fig2:
- Either expand on "outbreak" or provide a reference to the data/publication. All figures should be fully able to standalone and be interpreted without referring to the main body text.
- Check grammar and syntax in FastTree sentence, highlighted.
- References needed for tools used.
- Groups are indicated by coloured labels, but what does "d12 groups" stand for and what is "Support value"? Need to be defined for the less experienced user.

#### Raw data supplied
- Pipeline is fully available on Github but on line 157: 91 strains from Sierra Leone --where is this data? Is it publicly available? Also, data pertaining to what is referred as "Sanger" sequencing/genotyping in Table 1/ Table S1 is missing and should be included.

- Change the supplemental table to .csv/.tsv file format unless it is a requirement of PeerJ to produce supplementary data as an excel sheet.


Fig3:
- "identical spoligotyping patterns" was not stated or mentioned in main body text for these isolates. Lack of continuity with novel information provided in a table legend.

- Either expand on "outbreak" or reference the data source/publication. All figures should be fully able to standalone and be interpreted without referring to the main body text.

- If one isn't familiar with genomic distances and a distance matrix, it is unclear how the 4 isolates are "clearly separated". Suggest adding an explanatory sentence e.g. closely related strains are denoted by smaller pairwise distance values. Need to take audience into account if MTBseq is to be pitched as a diagnostic pipeline and accessible to all.

- "forming a tightly related cluster" --guessing this refers to the phylo tree?? It's out of context here and by what I can see on the tree, wrong?
1024-01, 3929-10, 6331-04 and 6821-03 do not form a tightly related cluster; 6821-03 appears as an outlier and 6631-04 separates from 1024-01 and 39239-10.

Experimental design

Kohl et al. present MTBseq, a pipeline which takes raw MTBC sequencing data, maps it to the reference H3Rv genome, detects variants associated with drug resistance and classifies the phylogenetic relationship between samples. They claim that the pipeline is customizable, expandable and high throughput and can be run on a laptop or desktop computer. They argue that a standardised pipeline is required for data analysis that is
accessible to all, regardless of their bioinformatic skills or specialised working environment.

### Pipeline issues
Line 122-123: "With MTBseq, we aim to provide a high throughput MTBC analysis
123 pipeline that is customizable, expandable, user friendly and, most importantly, standardized."

It is hard to see how MTBseq is high-throughput. The pipeline runs one sample at a time over a maximum of 8 threads and cannot be made high-throughput unless the user goes to a lot of effort to make it so. It goes against the ethos that is suggested throughout the paper regarding ease of use for those less experienced/ with no experience of bioinformatics. Suggest re-word.

With regards to MTBseq being customisable and expandable, I strongly feel that the authors should invest some time in highlighting examples and providing directions for this in their .readme file. Again, ease of use should be a priority here and at the moment there is very little/no direction for how one can change the 'standard' settings for each module in the pipeline or how to go about changing module entirely. For instance, can a user actually swap out the mapper and ensure both its success and compatibility with the rest of the pipeline?

With this in mind, I strongly encourage the authors to provide some toy data for their pipeline. This will not only allow less experienced users to navigate through each module/some of the modules of the pipeline and will provide invaluable information regarding input/output file formats. Refer to "Ten simple rules for making research software more robust", rule 9 (toy data)(http://journals.plos.org/ploscompbiol/article?id=10.1371/journal.pcbi.1005412).


I suggest improving checks on system return codes to allow for the pipeline to fail gracefully upon hitting an error. I ran the pipeline and allocated 10G of memory to process 3 toy MTB datasets. The pipeline stalled at the "indelrealigner" stage of the GATK module and I wouldn't have known that the process ran out of memory if the cluster had not notified me so. I could also not see any recommendations in the readme file regarding RAM usage. Strongly suggest fixing this bug. The pipeline ran well up until this error.


# .readme file
Needs to be improved and made more user friendly, especially for those not too familiar with bioinformatics. For example,
"You can ensure that MTBseq is executable from anywhere on your system by adding the MTBseq_source folder to your PATH variable or creating a symlink of MTBseq.pl to a folder that is already in your PATH variable."
would make little to no sense to an inexperienced user. Suggesting for the addition of example code.


- A note: Rule #4 from "Ten simple rules for making research software more robust" : Version your releases. I cannot see a --version flag defined in the readme file for MTBseq?

- usr/bin/perl should be replaced with usr/bin/env/ perl

- Typo (space) in "git clone https://github.com/ngs-fzb/MTBseq _source"

- States in the manuscript and readme for MTBseq that the reference genome can be changed to whatever bacterial reference you like. Some toy data highlighting this would be a useful addition.


- Line 213-215: Placement of this sentence here is odd - suggest switching it to methods.

- Line 124: "Therefore, all parameters are set to default values". What are these?
They should be specified in the manuscript/readme --can only be found in log files once pipeline has run.

- Why were the specific tools chosen over others? No clarity on this provided.

- Line 98: Define InDels

- Line 170: "Low frequency detection mode" has not been defined?

Validity of the findings

#### Research question well defined, relevant & meaningful. It is stated how the research fills an identified knowledge gap.
Kohl et al. present MTBseq, a pipeline which takes raw MTBC sequencing data, maps it to the reference H3Rv genome, detects variants associated with drug resistance and classifies the phylogenetic relationship between samples. They claim that the pipeline is customizable, expandable and high throughput and can be run on a laptop or desktop computer. They argue that a standardised pipeline is required for data analysis that is
accessible to all, regardless of their bioinformatic skills or specialised working environment.


#### Methods described with sufficient detail & information to replicate.
Stating that the MTBseq pipeline is "standardised" and yet fully customisable seems like an oxymoron to me. I strongly suggest re-wording this to reflect what I think the author means i.e. that the pipeline can either be run using default tool parameters or it can be customised by the user if required. "Standardised" suggests assurances that results will be the same regardless of platform used. This is not the case and in fact the authors allude to this themselves in the introduction (although too strongly) in that analysis results tend to vary between labs. Pipeline standardisation can be achieved using pipeline managers or containers (Docker, Snakemake, Nextflow).


- Line 171-172: Which are the 11 strains that could not be resolved by classical genotyping? These cannot be seen easily in Table S1?
- Line 180-181: Again, what are the thresholds for "unambiguous variant detection"? Need to be mentioned.

- Line 204-207: Please expand/speculate as to why this was the case.


- Line 229: "paving the way for efficient application of WGS"...where/how? Sentence is incomplete.

Additional comments

###Overall
I feel that MTBseq needs improvements in the areas outlined above prior to publication to ensure accessibility and ease of use for those implementing the pipeline in the intended audience.

Annotated reviews are not available for download in order to protect the identity of reviewers who chose to remain anonymous.

---

## Round 0.2 · Minor Revisions

A third reviewer with specific bioinformatics pipeline expertise was engaged after issues identified around the ease-of-use of MTBseq. Please carefully read and respond to the issues they've raised, in particular around installation.

Reviewer 1 ·

Basic reporting

In previous review, I requested that authors share the positions used for recalibration as a Supplemental Table in the manuscript. Authors argue it is sufficient to have this file on the GitHub repository and not in the manuscript. I disagree. Authors have use this list of resistance associated mutations when running MTBseq on the 91 strains from Sierra Leone and present sensitivity and specificity of resistance calls (vs. Sanger sequencing) based on these sites - therefore this data should be transparent and readily accessible to the reader as part of the publication. As authors express concern that it would be misleading to include these data, they can add a specific statement that their list of resistance-associated sites may not be comprehensive and that users are encouraged to produce their own preferred variant lists as desired.

I would like to thank the authors for clarifying the filtering process used by MTBseq and adding the relevant citation to Comas et al. 2010 for repetitive regions.

Line 114 - please provide the appropriate website for Picard (I recognize there is no citation for this but a link is needed)

Experimental design

Regarding comparing their pipeline to phenotypic DST - the authors' rationale makes sense to not include this. Thank you for the explanation. I might suggest removing the part about phenotypic data from the following statement, as it is not actually relevant to the aims herein and can potentially cause confusion: "The dataset used consisted of 91 well-characterized strains of a collection from Sierra Leone, for which both WGS data (ENA accession number PRJEB7727) and Sanger sequencing and phenotypic data was available."

Thank you also for adding cautionary comments regarding low frequency variants.

I thank the authors for providing some additional detail about the thresholds for variant calling in Section 3. However, I am concerned by the threshold of only 75% for calling a SNP. This is very low. If we have depth of 100x, this suggests it is acceptable to have up to 25 reads with the reference for this to be called a SNP - when in fact it suggests there may be a mixed population. While I recognize this is the default and can be altered, many users (unfortunately) will simply run this with default settings, taking this as a recommendation of best practice from a group with extensive TB genomics experience. Please increase this to a more confident threshold (e.g., 85-90%, as is found in other published genomics pipelines.

When authors state "variants need to be indicated by four reads mapped in forward and reverse direction" do they mean 2 forward reads and 2 reverse? Or 4 and 4? Please clarify this statement. I assume it is the former, because they then state "at least 4 calls with a P[capitalize]hred score of at least 20."

Please also clarify - are you using the BWA 'MEM' algorithm? This should be explicitly stated in the manuscript.

Validity of the findings

Thank you for the changes made in response to my comments here.

Additional comments

This pipeline is difficult to install and requires a lot of specific software and version numbers. Currently, this is not user-friendly (while I can respect the authors feel otherwise). While I can also appreciate that people who are non-bioinformaticians in the same institution as the authors have anecdotally been able to use this tool, this is not representative of the majority of institutions who would potentially see value in this tool. Please set up a bioconda or homebrew package to facilitate easier installation for users. This will make the tool much more accessible to others, which authors say is their intent, and this should be a minimum required for publication of this pipeline.

In relation to this, I would note that authors declined the other reviewer's request to present results in tab-delimited format because "the excel format has its advantages especially since both the article and MTBSeq software are not specifically targeted for bioinformaticians and programmers." This seems a bit incongruent with authors' expectation that the user must know basic linux to use their tool; if a user can run linux commands, it's not an unreasonable expectation that they should be able to open a tab-delimited file into a text editor - or Excel - based on their needs.

Reviewer 2 ·

Basic reporting

I am happy with the improvements that the authors have made with regards to grammar mistakes, sentence construction and overall readability of the manuscript. The corrections made to figures and supporting information are also overall acceptable. I am happy to see the mix-up with the publicly available datasets is corrected, that the Readme and error logs for the tool have been improved and that version control has been implemented. I still disagree with providing excel sheets rather than .csv/.tsv tables (which Excel will open with ease) as there are numerous formatting issues and errors that can occur (https://genomebiology.biomedcentral.com/articles/10.1186/s13059-016-1044-7). I recommend this work for publication.

Experimental design

no comment

Validity of the findings

no comment

Reviewer 3 ·

Basic reporting

no comment

Experimental design

no comment

Validity of the findings

no comment

Additional comments

I have assed only the software implementation and usability.

SUMMARY
=======

Code is not written by a software engineer, but is at a good standard.
No tests or testing scripts are included.
Would benefit from an example run through with data.
Needs to be packaged into Homebrew or Conda before publishing.

COMMENTS
========

Docs should be in Markdown not PDF

These are all core modules and don't need installing
- FindBin (v1.51)
- Cwd (v3.62)
- Getopt::Long (v2.5)
- File::Copy (v2.30)
- List::Util (v1.49)
- Exporter (v5.72)
- vars (v1.03)
- lib (v0.63)
- strict (v1.09)
- warnings (v1.34)

Why is it limited to 8 threads?
"MTBseq is able to run on 1 thread up to a maximum of 8 threads. "

"The re-compiled executables MUST be located within the appropriate folders:"
This is no acceptable, even via symlinks.
If the tools are in my PATH, it should use them.

Please don't distribute .tar.bz2 and untarred versions of the
samtools, bwa, picard and GATK tools

Are you legally allowed to distribute the GATK 3.8 JAR file?
I thought you needed to sign a click through licence?
And that licence is for personal use only?
If people install this on a cluster they may be violating the licence.

Your pipeline produces lots of intermediate files, inc many large BAMs.
Have you considered piping them together?

Much of the pipeline is just iterating over system() and print() commands.
Have you considered using a workflow system like Snakemake, Make, or Nextflow?

MTBseq.pl --version
Should print ONLY "MTBseq 1.0.1"

Author has written their own FASTA reader/writer, revcom and codon functions.
Did you consider using Bio::Perl or similar?

Lots of 'print $logprint || die' style lines.
This should be wrapped into a separate function.

Could implement your own mean() median() to avoid dependncy on
Statistics::Basic, but that would still leave the non-core MCE module.

The TBtools has masses of hash assignments for SNPs and lineages.
This would be much better configured as a separate config file,
to separate the implementation/code from the meta data.

Needs a --test or --check option to ensure everything is ready and working.
For example, I accidently set my BWA path wrong and got this:
".... 2>> /Users/tmp/tb/Bam/SAMPLE_LIB_FOO.bamlog failed: 32512"
Perhaps check at start of script that 'bwa' exists and works etc first.

Needs a walkthrough tutorial with some public data:
eg.
wget ftp://ena/tb/R1.fq.gz
wget ftp://end/tb/R2.fq.gz
MTB_seq.pl ....

It is not made clear which verion of Java is needed for GATK and Picard?
1.7, 1.8, 1.9 ?

The example command lines say "MTBseq" but I only have "MTBseq.pl" ?

It is not flexible in the filenames it will accept.
Our illumina instrument has _001 etc after the R1:
Sample18231_S81_R1_001.fastq.gz
It would be good to support an "--input_table" of the form
NAME <tab> READ1 <tab> READ2 <newline>
or similar.

I managed to run MTBfull with 4 threads ok on a single isolate.

It placed everything in the current directory.
This is not ideal behaviour in a command line tool.
Perhaps add an --outdir or similar option.

Most users will not want all the temporary files.
Perhaps have a --clean_up / --sauber_machen option to only keep final files.

END.

---

## Round 0.3 · accepted · Accept

I am satisfied with your responses.

#